# Effect of Borpolymer on Mechanical and Structural Parameters of Ultra-High Molecular Weight Polyethylene

**DOI:** 10.3390/nano11123398

**Published:** 2021-12-15

**Authors:** Sakhayana N. Danilova, Afanasy A. Dyakonov, Andrey P. Vasilev, Aitalina A. Okhlopkova, Aleksei G. Tuisov, Anatoly K. Kychkin, Aisen A. Kychkin

**Affiliations:** 1Department of Chemistry, Institute of Natural Sciences, North-Eastern Federal University, 677013 Yakutsk, Russia; afonya71185@mail.ru (A.A.D.); gtvap@mail.ru (A.P.V.); okhlopkova@ya.ru (A.A.O.); 2Institute of the Physical-Technical Problems of the North, Siberian Branch of the Russian Academy of Sciences, 677980 Yakutsk, Russia; kychkinplasma@mail.ru; 3Federal Research Centre “The Yakut Scientific Centre of the Siberian Branch of the Russian Academy of Sciences”, 677000 Yakutsk, Russia; tuisovag@gmail.com (A.G.T.); icen.kychkin@mail.ru (A.A.K.)

**Keywords:** ultra-high molecular weight polyethylene, polymethylene-p-triphenyl ester of boric acid, borpolymer, polymer composite materials

## Abstract

The paper presents the results of studying the effect of borpolymer (BP) on the mechanical properties, structure, and thermodynamic parameters of ultra-high molecular weight polyethylene (UHMWPE). Changes in the mechanical characteristics of polymer composites material (PCM) are confirmed and complemented by structural studies. X-ray crystallography (XRC), differential scanning calorimetry (DSC), scanning electron microscopy (SEM), and infrared spectroscopy (IR) were used to study the melting point, morphology and composition of the filler, which corresponds to the composition and data of the certificate of the synthesized BP. Tensile and compressive mechanical tests were carried out in accordance with generally accepted standards (ASTM). It is shown that BP is an effective modifier for UHMWPE, contributing to a significant increase in the deformation and strength characteristics of the composite: tensile strength of PCM by 56%, elongation at break by 28% and compressive strength at 10% strain by 65% compared to the initial UHMWPE, due to intensive changes in the supramolecular structure of the matrix. Structural studies revealed that BP does not chemically interact with UHMWPE, but due to its high adhesion to the polymer, it acts as a reinforcing filler. SEM was used to establish the formation of a spherulite supramolecular structure of polymer composites.

## 1. Introduction

Currently, polymer composite materials (PCM), due to their high mechanical properties and other special characteristics, low density, and ease of industrial processing, are widely used in industry, medicine, and other fields. Ultra-high molecular weight polyethylene (UHMWPE) is one of the promising polymers for manufacturing structural PCMs. It is known that UHMWPE is characterized by high chemical inertness, excellent mechanical properties, high impact strength and low coefficient of friction [1]. Due to these properties, UHMWPE is used, and can potentially be used in many areas from medicine to the space industry [2,3]. The introduction of micro- and nanosized fillers into UHMWPE increases the mechanical and tribological characteristics, which expands the application range of the material [4,5]. It is known that polymer composites based on UHMWPE filled with nanosized fillers are distinguished by a low coefficient of friction, increased strength characteristics, and resistance to cracking [6,7]. Due to the high specific surface area of particles and the decompensation of bonds of a significant number of atoms, nanosized fillers are characterized by their agglomeration, which leads to the appearance of defective regions and, consequently, to a decrease in the mechanical characteristics of PCM. There are studies on the modification of UHMWPE by the introduction of fibrous fillers [8,9], where there is an increase in the bearing capacity, wear resistance, rigidity, and strength of PCM [4]. The mechanical characteristics of fiber-filled composites depend on the interfacial interaction at the “fiber-polymer” interface, which requires additional modification of the fiber surface or the introduction of adhesion promoters into the PCM composition [10,11,12]. There are investigations in which polymers are used as a filler for UHMWPE. Such materials are characterized by increased wear resistance and low coefficient of friction [13,14,15,16], but at the same time they have low deformation and strength characteristics. For example, in the case such organic fillers as polyetheretherketones (PEEK) when creating composites based on UHMWPE, a decrease in mechanical parameters is shown [16]. A great number of studies are devoted to UHMWPE/PEEK composites and their wide application in the development of bone and hip implants, due to the biological characteristics of PEEK [17,18,19,20]. In [20], it was found that PEEK is poorly compatible with UHMWPE. However, a slight increase in the mechanical parameters of composites is explained by the high hardness of PEEK particles in comparison with UHMWPE [16].

In addition to the use of fillers to improve UHMWPE properties, inoculation techniques are used, including methods of ultrasonic treatment, mechanical activation, high-speed mixing of composite components, etc. [21,22,23]. In addition, specific methods of processing PCM are used, including the following: mixing a filler and polymer in solvents, adding surfactants [24,25,26], crosslinking UHMWPE macromolecules, modifying fillers by CVD—chemical vapor deposition [27], functionalizing fillers, etc. [28,29]. Despite a large number of publications on the study of the modification of UHMWPE composites components, the mechanisms for realizing the potential capabilities of PCM components have not yet been disclosed.

In this study, we investigated the effect of polymethylene-p-triphenyl ester of boric acid (BP) on the mechanical properties, structure, and thermodynamic parameters of UHMWPE, depending on its content. Polymethylene-p-triphenyl ester of boric acid is a class of organic boron compounds in which the B (boron) atom in the phenol molecule is linked through the O (oxygen) atom. Organic boron compounds are widely used in various fields: to increase the fire resistance of materials [30], to obtain porous materials [31], and as a polymer modifier [32,33,34]. There is a great deal of studies devoted to BP as an additive in epoxy resin and rubber [35,36,37,38,39,40]. However, borpolymers, in particular BP, as a filler for UHMWPE have not been investigated practically.

The aim of this research is to study the effect of boron polymer on the mechanical properties and structure of ultra-high molecular weight polyethylene. 

## 2. Materials and Methods

### 2.1. Materials and Obtaining of PCM

UHMWPE brand GUR-4022 (Celanese, Nanjing, China) was used as a polymer matrix, with a molecular weight of 5.0 × 10^6^ g/mol, a density of 0.93 g/cm^3^, and an average particle size of 145 μm. A synthesized polymethylene-p-triphenyl boric acid ester-PTBEC, called borpolymer (BP), was used as a modifying additive. BP was provided by Boroplast LLC (Boroplast, Biysk, Russia), with an average molecular weight of 2500–3000 a.u. and with a melting point of 150–160 °C.

To remove adsorbed moisture, the initial UHMWPE powder was preliminarily dried in a PE-0041 oven (Ekopribor, St. Petersburg, Russia) at a temperature of 85 °C for 1.5 h. UHMWPE and BP powder were mixed at room temperature in a paddle mixer with a rotor speed of 1200 rpm. The samples were prepared using the hot pressing technology in a PCMV-100 hydraulic vulcanization press (Impulse, Ivanovo, Russia) at a temperature of 175 °C, a pressure of 10 MPa, holding for 20 min and then cooling to room temperature. The borpolymer content in the polymer matrix was varied: 0.2, 0.3, 0.5, 1.0, 2.0, 3.0, and 5.0 wt. %.

### 2.2. Research Methods

The mechanical properties of UHMWPE and PCM were studied using the Autograph AGS-J tensile testing machine (Shimadzu, Tokyo, Japan). The tensile strength and elongation at break were tested according to ASTM D3039/D3039M-14 at the moving gripper speed of 50 mm/min, the number of samples was six. Compressive strength was determined according to ASTM D695.

X-ray diffraction patterns of the borpolymer and PCM was determine using X-ray powder diffractometry (XRD, ARL X’Tra, Thermo Fisher Scientific, Ecublens, Switzerland). An X-ray tube with a copper anode (λ (CuK_α_) = 0.154 nm) was used as a radiation source. For the study, we used samples in the form of plates with dimensions of 30 × 30 × 3 mm. The degree of crystallinity was determined by the Formula (1):(1)$$\alpha =\frac{A_{c}}{A_{c}+A_{a}}*100\%,$$
where *A_c_* is the area under the crystalline peaks, and *A_c_* + *A_a_* is the total area of both crystalline and amorphous regions. The average crystallite size (*L*) in the direction perpendicular to the crystal lattice plane was determined using the Scherrer Equation (2): (2)$$L=\frac{K\lambda }{\beta \cos\theta },$$
where *β* is the width at half maximum of the diffraction peak; *K* is the crystal lattice constant (approximately 0.9); λ—wavelength of the beam of monochromatic radiation CuK_α_, 0.154 nm; *θ* corresponds to the Bragg angle, and *L* corresponds to the average crystallite size. The distance (*d*) between the diffraction planes was obtained according to Bragg’s law (3):(3)2dsinθ=nλ,
where *n* is the diffraction order (integer); *d*—interplanar distance; *λ* is the wavelength of X-ray radiation, *θ* is the Bragg angle. 

The supramolecular structure of UHMWPE and PCM and powders of BP were studied on the JSM-7800F scanning electron microscope (Jeol, Akishima, Japan) with the X-MAX-20 attachment (Oxford Instruments plc, Tubney Woods, Abingdon, UK) in the secondary electron mode at an accelerating voltage of 1–1.5 kV.

The atomic force microscopy was performed with an NTEGRA instrument manufactured by NTegra Prima (NT-MTD, Zelenograd, Russia). The instrument was operated in ‘semi-contact mode’, which is often also referred to as “tapping mode”. Surface topography and phase images were obtained using NSG 10 golden silicone probes with a resonant frequency of 140–390 kHz and a force constant of 2.5–10 N/m. The AFM images obtained were processed using the “Nova” and “Image Analysis” software (NT-MTD, Zelenograd, Russia).

Fourier transform infrared (IR) spectroscopy (FTIR; Varian 7000, Palo Alto, CA, USA) was used to record IR spectra with an attenuated total reflection (ATR) attachment over the range 400–4000 cm^−1^.

The Raman spectra between 1600 cm^−1^ and 1000 cm^−1^ were recorded by using the NT-MDT NTEGRA (NT-MDT, Zelenograd, Russia) equipped in a 532 nm. The spectra were collected on three different points in one sample.

The thermodynamic characteristics of UHMWPE and composites were studied on a DSC 204 F1 Phoenix NETZSCH differential scanning calorimeter (Netzsch, Selb, Germany), where the measurement error was not more than ±0.1%, the heating rate was 20 °C/min, and the sample weight was 18 ± 1 mg. The measurements were carried out in a helium medium in a temperature range of 40–180 °C. The samples were placed in aluminum crucibles with a 40 µL. Temperature calibration was performed using standard samples of In, Sn, Bi, Pb, and KNO_3_. 

The degree of crystallinity of UHMWPE and PCM was calculated by the follow Equation (4):(4)$$\alpha ,\;\%=\frac{\Delta H_{endotherm}}{\Delta H_{f}\left(1-W_{f}\right)}\cdot 100\%,$$
where ∆*H_endotherm_*—is the melting enthalpy calculated from the area of endothermic melting peak; ∆*H_f_*—is the melting enthalpy for 100% crystalline UHMWPE, which is equal to 291 J/g; *W_f_*—is the mass content of the filler in PCM [1,41].

## 3. Results and Discussion

### 3.1. Characteristics of Borpolymer

Borpolymer belongs to the class of boric esters with a molecular weight distribution of the basic substance ≥99%. This substance is obtained by the polycondensation reaction of triphenyl ester of boric acid and 1, 3, 5—trioxane (paraformaldehyde) in an acidic medium.

Figure 1 shows the X-ray diffractogram of borpolymer. Based on the analysis of XRC diffraction patterns, it was established that the initial borpolymer is an amorphous compound, and a broad peak characteristic of amorphous compounds with a low intensity in the region (2θ = 10–30°) was found. No other peaks in the study area 2θ = 1.5–60° were found in the tested BP sample.

For a qualitative analysis of the BP composition, the BP structure was studied by IR spectroscopy (Figure 2).

As Figure 2 demonstrates, the IR spectrum shows the following characteristic peaks of BP benzene rings: 1045 and 1095 cm^−1^, corresponding to the vibrations of the C–H bond (methyl radical) in the plane of the benzene ring (in plane C–H blending), and a peak at 750 cm^−1^ due to vibration outside the plane of the benzene ring of the C–H bond of the methyl group (out of plane C–H blending). The peaks at 1590–1455 cm^−1^ correspond to the vibrations of the C=C bonds of the aromatic ring itself, and the intense absorption band in the region at 3290 cm^−1^ refers to the vibrations of the C–H bonds of the benzene ring [42]. The peaks of absorption bands of carbon-boron and oxygen-boron bonds were also found. Asymmetric stretching vibrations of the B–C bond in triphenylboron correspond to a peak at 1220 cm^−1^, while symmetric vibrations of the B–C bond are characterized by the occurrence of a peak at 825 cm^−1^. The peak at 1350 cm^−1^ is caused by bending vibrations of the B–O bond. Symmetric vibrations of this bond are marked by the occurrence of a low-intensity peak at 910 cm^−1^; possibly, the intensity decreases due to the strength of the B–O bond in the BP polymer [43]. The obtained IR spectra correspond to the chemical composition of BP.

Analysis of the DSC data indicates the presence of two melting peaks at 74 °C and 150 °C. The presence of two peaks indicates a polydisperse molecular weight distribution of the borpolymer, as the lower molecular weight portion of BP begins to melt at a relatively low temperature. The main melting peak on the DSC BP curve corresponds to 150 °C, and with its increase BP completely transforms into a molten state. It is known that BP is one of the promising heat-resistant additives for thermosetting plastics that increase the strength and wear resistance of materials [35,36,37,38]. Based on the temperature data, BP is suitable for the processing temperature range of UHMWPE based composites.

The sizes and morphology of the crushed BP particles were studied using a scanning electron microscope, the micrographs of which are shown in Figure 3.

The micrographs in Figure 3 show that the surface of the crushed particles is characterized by microdefects resulting from brittle fracture of BP. It was found that glass-like particles of BP are easily crushed; nevertheless, there is a wide variation in the size of crushed particles. It is noteworthy that the smaller BP particles are deposited on the surface of the larger ones. Obviously, at the stage of mixing the components of the polymer composition in a paddle mixer, due to mechanical effects, the BP particles will be dispersed with a fairly uniform distribution in the volume of the polymer.

### 3.2. Study of the PCM Structure

#### 3.2.1. IR Spectra of Composites

In order to determine the chemical effect of BP on the polymer matrix, the IR spectra of the initial UHMWPE and the UHMWPE/5 wt. % BP composite were studied (Figure 4). The IR spectra revealed the main peaks of UHMWPE at 2920, 2850, and 1470 cm^−1^, related to stretching and bending vibrations of -CH_2_ bonds and 1365 cm^−1^, corresponding to bending vibrations of -CH_3_. A crystallinity peak at 720 cm^−1^ was also found, due to pendulum vibrations of the polymer chain.

As can be seen from Figure 4, the IR spectrum of the UHMWPE/5 wt. % BP composite differs from the initial UHMWPE by the appearance of a broad absorption band in the 1260–1030 cm^−1^ region, which is characteristic of vibrations of the boron—oxygen bond, ether bond, or methyl group relative to the plane of the benzene ring. In addition, there is an insignificant peak at 1510 cm^−1^, which indicates the presence of carboxyl groups (C=O) or a C=C bond of the benzene ring of BP. It can be seen that the intensity of the detected peaks is minimal. It can be assumed that BP will not chemically interact with UHMWPE macromolecules and will not oxidize during PCM processing.

#### 3.2.2. Morphology of PCM

During the formation of composites and at the stages of processing, structural changes occur associated with a change in the supramolecular structure and the development of molecular orientation of polymer macromolecules. These changes in the structure of the polymer matrix determine the complex of mechanical properties of PCM. Due to the difference in the formation of the supramolecular structure, composites of the same polymer are often characterized by different values of mechanical parameters. Therefore, in order to determine the processes of structure formation in the supramolecular structure, UHMWPE and PCM conducted a SEM study, the results of which are shown in Figure 5.

Figure 5 shows that the supramolecular structure of the initial UHMWPE is characterized by a lamellar structure. The introduction of BP into UHMWPE transforms the lamellar structure into a spherulite structure of the radial type with irregularly shaped elements. Composite with 0.5 wt. % BP is characterized by the formation of large spherulites with clearly defined boundaries. In the case of 1 wt. % BP in UHMWPE, a decrease in the size of spherulite structures is observed. The supramolecular structure of the composite containing 2 and 5 wt. % filler becomes more disordered, defect regions are recorded, which will further affect the mechanical properties of the material. At the same time, these composites contain fan-shaped spherulites.

#### 3.2.3. Investigation of the Structure of Composites by the AFM Method

Structural studies of the composites were carried out using the AFM method in a semicontact mode, which makes it possible to obtain a high contrast in the visualization of submicron structures and to recognize various components in heterogeneous polymer systems (Figure 6). In the case of a smooth but chemically dissimilar surface, it is possible to visualize surface areas that differ in phase composition. Since the detection of the oscillation phase occurs simultaneously with the acquisition of the surface topography with the amplitude detection of the probe position in the feedback, it is possible to obtain information on the phase composition of the sample from the comparison of the amplitude and phase images. In this work, the object of study was a composite based on UHMWPE and BP, where the latter particles act as a dispersed phase. Therefore, the phase-contrast on the AFM made it possible to estimate the degree of BP distribution in the volume of the matrix and to measure the size of the crushed filler particles during the processing of the composite [44].

Figure 6 shows 3D images of the topography and phase-contrast of the composite slice containing 0.5 wt. % BP. The choice of this composition of the composite for research on AFM is due to its better mechanical properties. The scanning area was 1 × 1 μm. It was found that the distribution of BP particles in the matrix volume is chaotic. Phase-contrast analysis revealed the presence of a scatter in the sizes of BP particles (from 8.5 nm to ~70 nm). In this case, small BP particles form agglomerates, which, upon crystallization of UHMWPE, orient the crystal growth with the formation of spherulites, where they act as crystallization centers. It was registered that some part of nanosized BP particles are concentrated along the boundaries of spherulite formations due to their migration during pressing. It is known that if a multicomponent material contains several different (non-gaseous) phases, in which at least one of the phases has at least one dimension of the order of nanometers, then it belongs to nanomaterials. Thus, we have shown the formation of a nanocomposite upon the introduction of nanosized BP particles into UHMWPE.

#### 3.2.4. XRC of Composites

Structural studies of UHMWPE and PCM were carried out by X-ray structural analysis (Figure 7). From the X-ray diffraction patterns of all samples, two obvious intense peaks at 2θ ≈ 21.5° and 24.0° can be distinguished, corresponding to the crystallographic planes (110) and (200) of the UHMWPE polymer [45]; no other peaks were found. The original BP is an amorphous compound as noted above. When BP was injected into UHMWPE, no additional peaks were recorded on PCM radiographs.

Table 1 shows the results of XRD analysis of UHMWPE and UHMWPE/BP composites.

As Table 1 suggests, the introduction of a borpolymer into UHMWPE reduces the degree of crystallinity by 3% at a filler content from 0.2 to 3 wt. %, calculated from the ratio of the intensities of the crystalline and amorphous phases. The degree of crystallinity of the UHMWPE/5 wt. % BP composite decreased by 14% relative to the initial polymer. This may be due to the effect of agglomeration of the filler, which limits the molecular mobility of polymer chains and prevents the crystallization of the polymer [46]. The crystallite sizes of PCM, calculated according to the Scherrer equation at a content of 0.2 to 1 wt. % BP, remain at the level of the initial polymer; with a further increase in the BP content from 2 to 5 wt. % in UHMWPE, a decrease in the crystallite size is observed.

#### 3.2.5. Raman Spectra of Composites

Figure 8 shows the Raman spectra of the initial UHMWPE and the composite containing 5 wt. % BP. Raman spectra are sensitive to vibrations of the crystal lattice (crystalline state) of polyethylene; due to this, these spectra are used to explain the effect of fillers on the phase state of the matrix [47]. In this case, vibrational absorption bands are recorded in the region of 1000 and 1600 cm^−1^, caused by the twisting of the δ(CH_2_) bond and the stretching of the bonds (CC).

As can be seen from Figure 8, in the Raman spectrum of UHMWPE and PCM, characteristic peaks in the region of 1060 and 1123 cm^−1^ are visible, referring to symmetric and asymmetric stretching vibrations of the C–C bond in the crystalline phase of PE. The peak at 1292 cm^−1^ corresponds to the bending torsional vibrations of the CH_2_ group in the crystalline phase. The absorption bands in the region of 1440 and 1461 cm^−1^ refer to bending vibrations of the CH_2_ group of the amorphous phase of polyethylene [48,49]. In the Raman spectrum, the indicator of crystallinity of polyethylene is a peak at 1416 cm^−1^, which is weakly expressed in the initial UHMWPE and PCM. It was found that the introduction of BP into UHMWPE leads to broadening of the absorption band related to vibrations in the amorphous phase. There is also a decrease in the intensity of the 1292 cm^−1^ peak of the τCH_2_ crystalline vibration. The results obtained indicate a decrease in the crystallinity of UHMWPE upon the introduction of BP, and, on the whole, agree with the results of X-ray diffraction analysis. Thus, the introduction of BP into UHMWPE leads to a decrease in the crystallinity of the composite.

### 3.3. Thermodynamic Properties of PCM

Figure 9 shows the DSC data curves obtained by heating the samples, and Table 2 presents the data of the study results.

As evident from Figure 9 and Table 2, the temperature of the onset of melting of PCM does not change over the entire concentration range. Some shift of the melting peaks is observed, but these changes are insignificant and are included in the measurement error range. There is a narrowing of the DSC curves of the composites in comparison with the original UHMWPE.

It was shown that the degree of crystallinity of the initial UHMWPE is 58.7%. After the introduction of BP, a decrease in the degree of crystallinity by 18% is observed. In general, the degree of crystallinity of composites in the entire concentration range of filling is 47–48%. DSC crystallinity values differ from XRC data. However, both methods demonstrate a similar change trend, which is associated with the amorphization of UHMWPE with the introduction of BP, which leads to a decrease in the degree of crystallinity. Thus, the filler affects the growth and shape of crystallites in the process of PCM structuring, which consists in some deformation of the crystalline regions [50].

It was found that the enthalpy of melting of the composites decreases in comparison with the initial UHMWPE. In a series of composites, the enthalpy of melting gradually decreases with an increase in the BP content, which is associated with the loosening of the UHMWPE structure. The authors in [41] argue that the decrease in enthalpy is caused by the nature of the interaction in the compositional system. If the interaction between the polymer and the filler prevails, where the active surface of the filler acts as a nucleating agent (to heterogeneous nucleation) during crystallization, this leads to an increase in the enthalpy of melting and the degree of crystallinity. This tendency is observed in heterogeneous systems, where the filler has a high surface activity [51,52,53,54]. In the case of a predominant interaction between filler particles, the formation of agglomerates is observed, which limits the rate of polymer crystallization. Thus, it was shown that with an increase in the BP content in PCM, agglomeration between filler particles intensifies, which leads to a decrease in the enthalpy of melting by 20% compared to the initial UHMWPE. In this case, the formation of less perfect and defective structural elements—spherulites is observed in the supramolecular structure of PCM (Figure 5d,e). Nevertheless, a decrease in the degree of crystallinity and enthalpy of melting does not lead to a deterioration in the mechanical properties of the composite. It is known that the amorphous phase in UHMWPE contributes to an increase in the impact toughness of the material, due to the effect of linkage of through-feed chains [55].

Thus, the introduction of BP contributes to an overall decrease in the degree of crystallinity and the enthalpy of fusion of UHMWPE.

### 3.4. Mechanical Properties of PCM

BP is actively used as a hardening modifier for thermosetting plastics and industrial rubber goods, which explains the increased interest in this material. The mechanical characteristics of UHMWPE filled by BP are presented in Table 3 and Figure 10.

For the stress–strain curve, we took the test data of composites, which are similar to the average values after statistic processing.

Analysis of the results of PCM mechanical characteristics showed that 0.2 and 0.5 wt. % BP content leads to a significant increase in the strength and elasticity of the material. The increase in tensile strength of PCM was noted by 53% and 56% relative to the original polymer, respectively. At the same time, there is an increase in the elongation at break by 28% and 23%. A further increase in BP content leads to a gradual decrease in these parameters. However, the value of the elongation at break of the composite containing 5 wt. % BP, remains within the measurement error. The tensile strength of the UHMWPE/5 wt. % BP composite is 18% higher compared to unfilled UHMWPE. The modulus of elasticity of the composites and the original UHMWPE does not undergo significant changes, which indicates that the rigidity of the material is preserved throughout the entire concentration range of filling.

Based on studies of the supramolecular structure of PCM, it was found that the introduction of low concentrations of BP forms a fine-spherulite structure, which explains the maximum increase in mechanical parameters. At high concentrations, the occurrence of defective areas is observed, which leads to a slight decrease in mechanical parameters relative to PCM with a lower BP content, and does not decrease as compared to the original UHMWPE.

Studies on the modification of the UHMWPE matrix with thermosetting polymers or organic compounds of the ester class are poorly understood. In addition, the use of this borpolymer as a thermoplastic modifier has not been previously considered. Wang et al. showed the effect of thermosetting polymers on the wettability of UHMWPE fibers [56,57]. It was found that thermosetting binders of various types increase the wettability of UHMWPE fibers, thereby increasing the strength of the material by enhancing the adhesive interaction between the components of the composite. In [11] it was found that polyphenyl ether combined with carbon fibers increases the wear resistance of PCM, also due to the enhancement of interfacial interaction between the components, and due to the participation of ether in the formation of secondary structures on the friction surface. BP is known to be used effectively in elastomeric materials as a reinforcing agent. In this case, BP acts as a modifier of the rubber matrix, contributing to the formation of a stable three-dimensional vulcanization network. Moreover, the presence of a boron atom enhances the interaction at the polymer–filler interface, which indicates the reactivity of BP during vulcanization [35]. However, the results of IR spectroscopy of the UHMWPE/5 wt. % BP composite (Figure 4) indicate that the filler particles do not interact with the polymer macromolecule. Thus, BP acts as a reinforcing modifier for the polymer matrix.

In addition to the effect of strengthening the polymer matrix, an increase in the deformation and strength characteristics is due to the formation of the spherulite structure of PCM [4,58]. It is known [59] that PCMs with small spherulites are usually more rigid than composites consisting of large spherulites (Figure 5). Mechanical deformation of composites with a spherulite structure first destroys the boundary regions of the spherulites, i.e., the interlamellar amorphous part. Then the inner part of the spherulites undergoes deformation, since the crystalline ordered phase of the polymer is stronger [60]. Thus, composites characterized by a large amount of spherulites, for example, in a fine-spherulite structure, will have increased strength, while in composites characterized by the formation of an inhomogeneous and coarse-spherulite structure, the boundary regions are usually weak [58]; therefore, with an increase in the filler content, the occurrence of defective regions in the supramolecular structure of PCM is observed, which is accompanied by a slight decrease in mechanical parameters (Figure 5d,e). In [61], data are provided showing that organic fillers with a low molecular weight plasticize the UHMWPE matrix during stretching, facilitating relaxation processes. The increase in the relative elongation of composites containing BP can be explained by the plasticizing effect of BP.

The results of studying the effect of borpolymer on the compressive strength of PCM at different relative deformations are presented in Table 4 and Figure 11.

It was found that the introduction of BP into the polymer leads to an increase in compressive stress, at a specified relative deformation of 2.5%, by about 2–3 times compared with the initial UHMWPE. High values of the compressive stress at a specified relative deformation of 10% are observed for the composite with the composition UHMWPE/0.2 wt. % BP and UHMWPE/2 wt. % BP, in which an increase of 65% and 53% is noted, respectively. The compressive stress, at compressive strength at 25% strain of the composites, changes insignificantly depending on the filler content and remains within the measurement error. The increase in compressive strength values is attributed to an increase in the material’s resistance to deformation during compression, due to the formation of a reinforced PCM system [62]. In addition, it is assumed that, due to the high molecular weight of UHMWPE, regions with large overlaps of long macromolecule chains are formed. The occurrence of such zones with large overlap increases the ability of PCM to transfer a large compressive force from molecule to molecule [63].

Thus, the introduction of BP into the polymer leads to an increase in the deformation-strength characteristics and compressive strength, even at a low filler content.

## 4. Conclusions

The effect of borpolymer on the mechanical properties and structure of UHMWPE has been studied. It was found that the use of borpolymer as a UHMWPE modifier made it possible to increase the mechanical characteristics of the material at low BP concentrations (at 0.2 and 0.5 wt. %). At these concentrations, a maximum increase in tensile strength of 56% and elongation at break of 28%, relative to the original UHMWPE, was recorded. An increase in compressive strength was established at a specified relative deformation of 2.5% and 10% in the entire concentration range of PCM; the maximum values of these indicators were 13 MPa and 28 MPa, respectively. No significant changes in the modulus of elasticity are observed. The study of the processes of structure formation by the SEM method revealed the formation of a spherulite structure upon the introduction of BP, which explains the increase in the tensile strength of PCM. By means of IR spectroscopy, it was found that the borpolymer does not enter into chemical interactions with UHMWPE during processing. The presence of the main peaks of absorption caused by the vibrations of bonds of the initial components UHMWPE and BP was found. DSC and XRC studies of the degree of crystallinity revealed a general decrease in this parameter caused by loosening and amorphization of the structure with increasing BP concentration. These changes lead to a decrease in the enthalpy of melting by 20% compared to the initial polymer. The increase in the elasticity of the material is explained by the fact that the introduction of an amorphous filler into UHMWPE facilitates relaxation processes when an external load is applied.

Thus, BP is an effective filler for UHMWPE, helping to increase the tensile strength and elongation of a composite.

## Figures and Tables

**Figure 1 nanomaterials-11-03398-f001:**
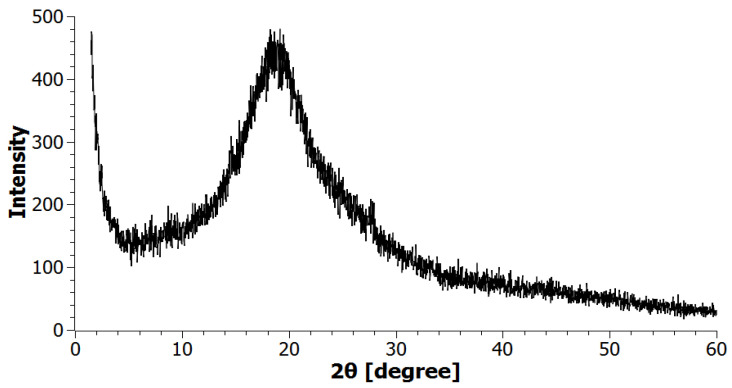
XRD pattern of raw BP.

**Figure 2 nanomaterials-11-03398-f002:**
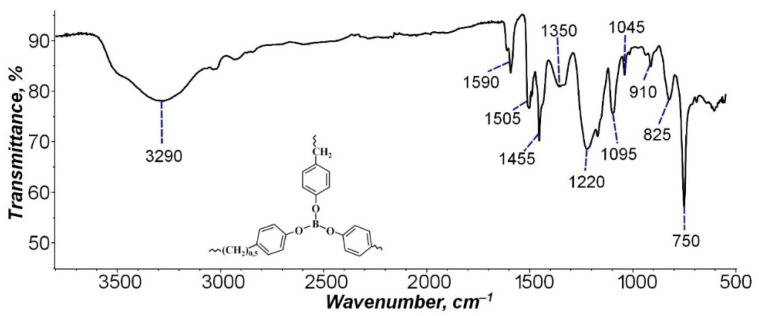
IR spectra of borpolymer.

**Figure 3 nanomaterials-11-03398-f003:**
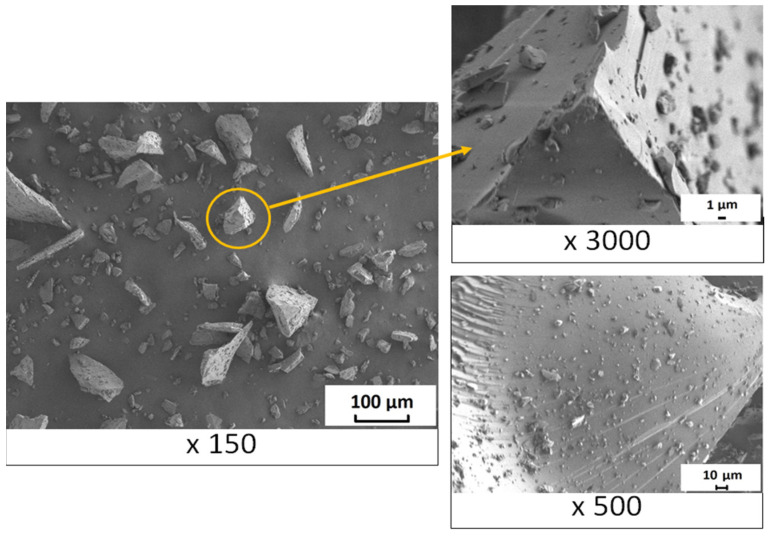
Micrographs of BP particles.

**Figure 4 nanomaterials-11-03398-f004:**
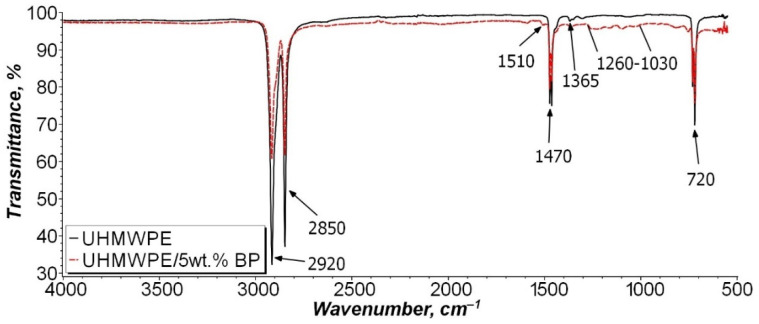
IR spectra of the initial UHMWPE and composite UHMWPE/5 wt. % BP.

**Figure 5 nanomaterials-11-03398-f005:**
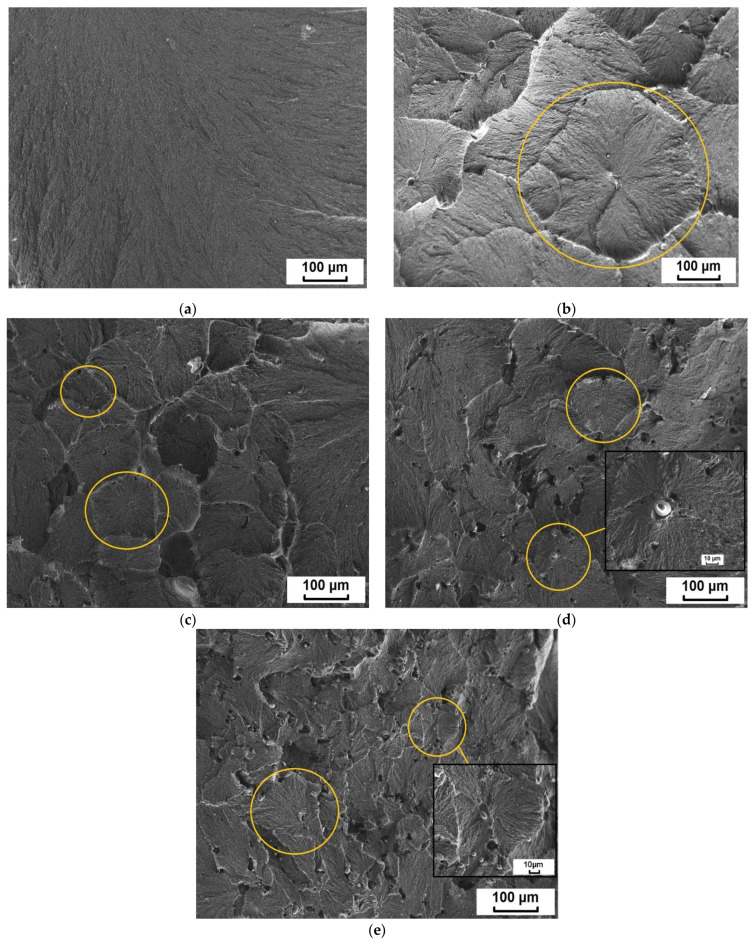
Microphotographs of the structure of (**a**) the initial ultra-high-molecular weight polyethylene (UHMWPE) and polymer composite materials (PCMs) based on UHMWPE filed by BP (**b**) 0.5 wt. %, (**c**) 1.0 wt. %, (**d**) 2.0 wt. % and (**e**) 5.0 wt. %.

**Figure 6 nanomaterials-11-03398-f006:**
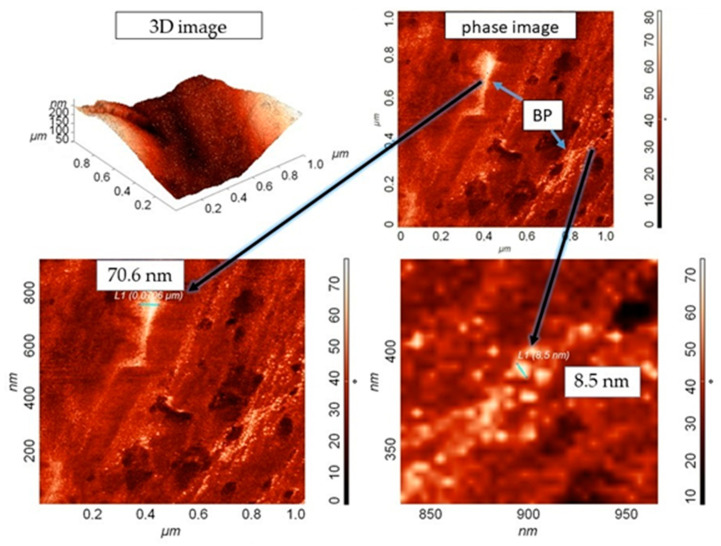
AFM 3D image of topography and phase-contrast of the composite slice, containing 0.5 wt. % BP.

**Figure 7 nanomaterials-11-03398-f007:**
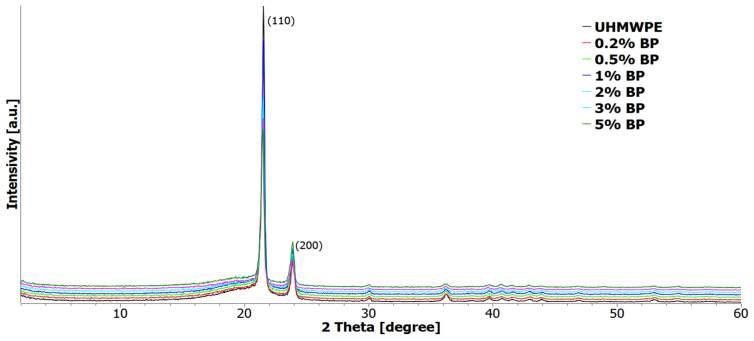
X-ray diffraction patterns of UHMWPE and PCM.

**Figure 8 nanomaterials-11-03398-f008:**
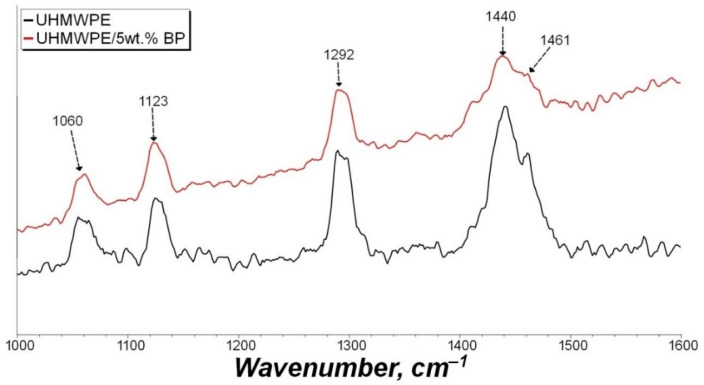
Raman spectra of the initial UHMWPE and composite UHMWPE/5 wt. % BP.

**Figure 9 nanomaterials-11-03398-f009:**
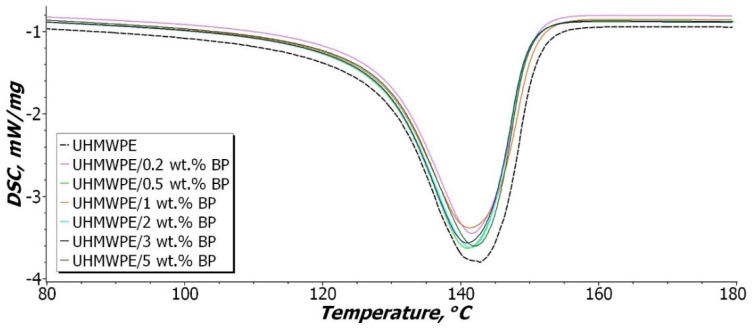
Heating melting function curve of UHMWPE and PCM.

**Figure 10 nanomaterials-11-03398-f010:**
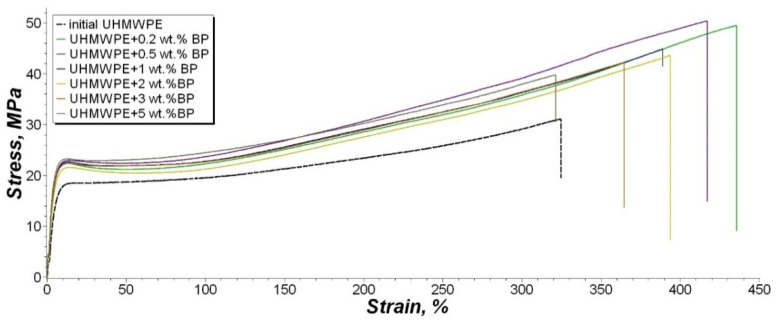
Stress–strain curve of the tensile tests.

**Figure 11 nanomaterials-11-03398-f011:**
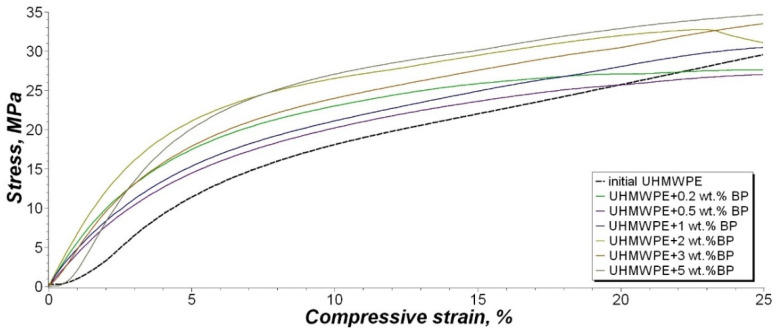
Stress–strain curve of the compressive tests.

**Table 1 nanomaterials-11-03398-t001:** Results of X-ray structural analysis.

Samples	X-ray Structural Analysis			
	α, %	2θ (°)	L, nm	d, nm
initial UHMWPE	58	21.5096	34.15	0.41
UHMWPE + 0.2% BP	56	21.4988	33.35	0.41
UHMWPE + 0.5% BP	55	21.4938	33.17	0.41
UHMWPE + 1% BP	56	21.4933	34.41	0.41
UHMWPE + 2% BP	56	21.4640	32.39	0.41
UHMWPE + 3% BP	56	21.4394	31.75	0.41
UHMWPE + 5% BP	50	21.4547	31.83	0.41

Notes: α—degree of crystallinity, %; 2θ—angle θ, (°); L—crystallite size, nm; d—interplanar distance, nm.

**Table 2 nanomaterials-11-03398-t002:** Melting point, melting enthalpy, and degree of crystallinity of UHMWPE and composite.

Samples	Thermodynamic Properties		
	T_onset_, °C	ΔH_me_, J/g	α, %
initial UHMWPE	127.7	171.1	58.7
UHMWPE + 0.2% BP	128.4	139.3	47.9
UHMWPE + 0.5% BP	128.0	138.5	47.8
UHMWPE + 1% BP	127.6	137.2	47.6
UHMWPE + 2% BP	128.1	138.4	48.5
UHMWPE + 3% BP	127.9	135.7	48.1
UHMWPE + 5% BP	128.4	135.2	48.9

Notes: T_onset_—melting point onset temperature, °C; ΔH_me_—melting enthalpy, J/g; α—degree of crystallinity, %.

**Table 3 nanomaterials-11-03398-t003:** Elongation at break, tensile strength, and Young’s modulus of UHMWPE and PCM with borpolymer (BP).

Samples	σ_T_, MPa	ε_b_, %	E, MPa
initial UHMWPE	32 ± 3	339 ± 16	420 ± 26
UHMWPE + 0.2% BP	49 ± 1	434 ± 14	472 ± 34
UHMWPE + 0.5% BP	50 ± 1	417 ± 10	524 ± 37
UHMWPE + 1% BP	45 ± 1	389 ± 8	499 ± 19
UHMWPE + 2% BP	43 ± 2	383 ± 18	519 ± 32
UHMWPE + 3% BP	43 ± 1	369 ± 12	524 ± 21
UHMWPE + 5% BP	39 ± 1	327 ± 14	520 ± 22

Notes: σ_T_—tensile strength, MPa; ε_b_—elongation at break, %; E—Young’s modulus in deformation 0.1–0.3%, MPa.

**Table 4 nanomaterials-11-03398-t004:** Melting point, melting enthalpy, and degree of crystallinity of UHMWPE and composite.

Samples	Compressive Strength		
	2.5% Deformation	10% Deformation	25% Deformation
initial UHMWPE	4 ± 1	17 ± 2	30 ± 1
UHMWPE + 0.2% BP	11 ± 2	28 ± 2	29 ± 2
UHMWPE + 0.5% BP	9 ± 1	21 ± 1	26 ± 2
UHMWPE + 1% BP	9 ± 2	23 ± 1	30 ± 1
UHMWPE + 2% BP	13 ± 2	26 ± 2	31 ± 1
UHMWPE + 3% BP	11 ± 1	24 ± 3	33 ± 1
UHMWPE + 5% BP	10 ± 2	25 ± 1	34 ± 1

## Data Availability

The data used to support the findings of this study are available from the corresponding author upon request.

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
