# Peer review of "Effect of Borpolymer on Mechanical and Structural Parameters of Ultra-High Molecular Weight Polyethylene"

_nanomaterials, 2021, doi:10.3390/nano11123398_

Round 1
Reviewer 1 Report
In this manuscript, boron polymer was added into UHMWPE, and the effect of boron polymer on the mechanical properties, structure and thermodynamic parameters of UHMWPE was investigated. The results showed a great increase of the tensile strength, elongation, and compressive strength of UHMWPE by adding boron polymer. The manuscript is important for the improvement of mechanical properties of UHMWPE. This manuscript was good presented in results. But the mechanism is confusing, boron polymer and UHMWPE is incompatible, and there is not any special interaction between them, why boron polymer uniformly dispersed and improve the mechanism properties of UHMWPE? This should be taken into consideration by the authors to improve the quality of manuscript.
Author Response
Response to Reviewer 1 Comments
In this manuscript, boron polymer was added into UHMWPE, and the effect of boron polymer on the mechanical properties, structure and thermodynamic parameters of UHMWPE was investigated. The results showed a great increase of the tensile strength, elongation, and compressive strength of UHMWPE by adding boron polymer. The manuscript is important for the improvement of mechanical properties of UHMWPE. This manuscript was good presented in results. But the mechanism is confusing, boron polymer and UHMWPE is incompatible, and there is not any special interaction between them, why boron polymer uniformly dispersed and improve the mechanism properties of UHMWPE? This should be taken into consideration by the authors to improve the quality of manuscript.
Response to comment: Thank you for your comment! The article stated that the borpolymer acts as a reinforcing filler for UHMWPE (line 402). Uniform dispersion of the particles is achieved by mixing in a paddle mixer (line 95-96). In addition, borpolymer particles contribute to forming a spherulitic structure, which explains the increase in mechanical parameters (lines 332-340).
Reviewer 2 Report
UHMWPE with different content of BP, whether the structure and crystal shape can be characterised by Raman spectra?Please have a try.
Author Response
Response to Reviewer 2 Comments
UHMWPE with different content of BP, whether the structure and crystal shape can be characterised by Raman spectra?Please have a try.
Response to comment: Thank you for your comment! Discussion of Raman spectra added to the article.
Reviewer 3 Report
In the manuscript all necessary information is captured by 7 figures and 2 tables. There are 59 references, all of them are adequate and are reflected in the text except reference [58].
All studies are conducted at a high scientific level and the manuscript is well written. The obtained results are important for understanding the processes that occur in the blend of UHMWPE with boron polymer (BP). The differences in the blend are very small except the mechanical properties. The stress-strain graphs of the tensile and compressive tests should be included as well.
Overall the paper is sufficiently presented and the results are interesting although the novelty is not the highest. The described manuscript is comprehensive, however it is not clear how the blend of polymers corresponds to the field of Nanomaterials.
The authors should provide sufficient evidence that their work is relevant with the scope of the journal.
Author Response
Response to Reviewer 3 Comments
Comment #1: In the manuscript all necessary information is captured by 7 figures and 2 tables. There are 59 references, all of them are adequate and are reflected in the text except reference [58].
Response to comment #1: Thank you for your comment! Note corrected.
Comment #2: All studies are conducted at a high scientific level and the manuscript is well written. The obtained results are important for understanding the processes that occur in the blend of UHMWPE with boron polymer (BP). The differences in the blend are very small except the mechanical properties. The stress-strain graphs of the tensile and tests should be included as well.
Response to comment #2: Thank you for your comment! The work proved that the introduction of a small amount of filler leads to a significant increase in the mechanical properties of the composite. At high concentrations, the filler begins to agglomerate (line 262, 287, 345); therefore, it is not advisable to investigate highly filled composites. The stress-strain graphs of the tensile and compressive tests were added.
Comment #3: Overall the paper is sufficiently presented and the results are interesting although the novelty is not the highest. The described manuscript is comprehensive; however, it is not clear how the blend of polymers corresponds to the field of Nanomaterials.
Response to comment #3: Thank you for your comment! The novelty of the work is the study of the effect of borpolymer on the properties and structure of UHMWPE, and such studies have not been conducted before. It is established that modification of UHMWPE by borpolymer leads to the formation of a nanocomposite with a specific structural organization that is not characteristic of the original UHMWPE (Fig.5). The transformation of the supramolecular structure of UHMWPE into a spherulite one with the formation of structural elements of various geometric shapes with well-defined boundaries in which borpolymer nanoparticles act as crystallization centers is registered. With increased filler content, the localization of borpolymer particles in the boundary regions is established. Numerous works are known in which such a transformation of the structural organization of polymers with the introduction of nanoscale fillers is defined as the formation of a nanocomposite [10.15406/mojabb.2017.01.00030, 10.1063/1.5017337, 10.1016/j.compositesb.2019.107181]. In this regard, the studied composites are classified as nanomaterials.
Comment #4: The authors should provide sufficient evidence that their work is relevant with the scope of the journal.
Response to comment #4: Thank you for your comment! The composites based on UHMWPE and borpolymer studied in this work are classified as nanomaterials by their structural parameters. To prove this, the article introduces the results of a study of the structural features of the composite occurring at the nanoscale by the AFM method. The formation of a multiphase system in the PCM, characterized by phase contrast, localization of nanoscale BP particles along the boundaries of spherulites, has been established. A discussion of AFM figures and their analysis has been added to the article.

Round 2
Reviewer 3 Report
The authors have addressed all the comments. The manuscript can be published.